# Metabolic Dysregulation Explains the Diverse Impacts of Obesity in Males and Females with Gastrointestinal Cancers

**DOI:** 10.3390/ijms241310847

**Published:** 2023-06-29

**Authors:** Spencer R. Rosario, Bowen Dong, Yali Zhang, Hua-Hsin Hsiao, Emily Isenhart, Jianmin Wang, Erin M. Siegel, Arta M. Monjazeb, Dwight H. Owen, Prasenjit Dey, Fred K. Tabung, Daniel J. Spakowicz, William J. Murphy, Stephen Edge, Sai Yendamuri, Sami Ibrahimi, Jill M. Kolesar, Patsy H. McDonald, Deepak Vadehra, Michelle Churchman, Song Liu, Pawel Kalinski, Sarbajit Mukherjee

**Affiliations:** 1Department of Biostatistics and Bioinformatics, Roswell Park Comprehensive Cancer Center, Buffalo, NY 14203, USA; spencer.rosario@roswellpark.org (S.R.R.); yali.zhang@roswellpark.org (Y.Z.); hua-hsin.hsiao@roswellpark.org (H.-H.H.); emily.isenhart@roswellpark.org (E.I.); jianmin.wang@roswellpark.org (J.W.); song.liu@roswellpark.org (S.L.); 2Department of Pharmacology and Therapeutics, Roswell Park Comprehensive Cancer Center, Buffalo, NY 14203, USA; 3Department of Immunology, Roswell Park Comprehensive Cancer Center, Buffalo, NY 14203, USA; bowen.dong@roswellpark.org (B.D.); prasenjit.dey@roswellpark.org (P.D.); pawel.kalinsky@roswellpark.org (P.K.); 4Department of Cancer Epidemiology, H. Lee Moffitt Cancer Center, Tampa, FL 33612, USA; erin.siegel@moffitt.org; 5Department of Radiation Oncology, University of California Davis, Sacramento, CA 95616, USA; ammonjazeb@ucdavis.edu; 6Department of Medical Oncology, Ohio State University Comprehensive Cancer Center, Columbus, OH 43210, USA; dwight.owen@osumc.edu (D.H.O.); daniel.spakowicz@osumc.edu (D.J.S.); 7Department of Epidemiology, Ohio State University Comprehensive Cancer Center, Columbus, OH 43210, USA; fred.tabung@osumc.edu; 8Department of Immunology, University of California Davis, Sacramento, CA 95616, USA; wmjmurphy@ucdavis.edu; 9Department of Surgical Oncology, Roswell Park Comprehensive Cancer Center, Buffalo, NY 14203, USA; stephen.edge@roswellpark.org; 10Department of Thoracic Surgery, Roswell Park Comprehensive Cancer Center, Buffalo, NY 14203, USA; sai.yendamuri@roswellpark.org; 11Department of Medicine, Oklahoma University Health Stephenson Cancer Center, Oklahoma City, OK 73104, USA; sami-ibrahimi@ouhsc.edu; 12Department of Pharmacy, University of Kentucky College of Pharmacy, Lexington, KY 40506, USA; jill.kolesar@uky.edu; 13Department of Cancer Biology, Moffitt Cancer Center, Tampa, FL 33612, USA; patsy.mcdonald@moffitt.org; 14Department of Medicine, Roswell Park Comprehensive Cancer Center, Buffalo, NY 14203, USA; deepak.vadehra@roswellpark.org; 15Precision Therapy and Diagnostics, Aster Insights, Hudson, FL 34667, USA; michelle.churchman@asterinsights.com

**Keywords:** obesity, metabolism, cancer, omics, immunity, gender disparity

## Abstract

The prevalence of obesity, defined as the body mass index (BMI) ≥ 30 kg/m^2^, has reached epidemic levels. Obesity is associated with an increased risk of various cancers, including gastrointestinal ones. Recent evidence has suggested that obesity disproportionately impacts males and females with cancer, resulting in varied transcriptional and metabolic dysregulation. This study aimed to elucidate the differences in the metabolic milieu of adenocarcinomas of the gastrointestinal (GI) tract both related and unrelated to sex in obesity. To demonstrate these obesity and sex-related effects, we utilized three primary data sources: serum metabolomics from obese and non-obese patients assessed via the Biocrates MxP Quant 500 mass spectrometry-based kit, the ORIEN tumor RNA-sequencing data for all adenocarcinoma cases to assess the impacts of obesity, and publicly available TCGA transcriptional analysis to assess GI cancers and sex-related differences in GI cancers specifically. We applied and integrated our unique transcriptional metabolic pipeline in combination with our metabolomics data to reveal how obesity and sex can dictate differential metabolism in patients. Differentially expressed genes (DEG) analysis of ORIEN obese adenocarcinoma as compared to normal-weight adenocarcinoma patients resulted in large-scale transcriptional reprogramming (4029 DEGs, adj. *p* < 0.05 and |logFC| > 0.58). Gene Set Enrichment and metabolic pipeline analysis showed genes enriched for pathways relating to immunity (inflammation, and CD40 signaling, among others) and metabolism. Specifically, we found alterations to steroid metabolism and tryptophan/kynurenine metabolism in obese patients, both of which are highly associated with disease severity and immune cell dysfunction. These findings were further confirmed using the TCGA colorectal adenocarcinoma (CRC) and esophageal adenocarcinoma (ESCA) data, which showed similar patterns of increased tryptophan catabolism for kynurenine production in obese patients. These patients further showed disparate alterations between males and females when comparing obese to non-obese patient populations. Alterations to immune and metabolic pathways were validated in six patients (two obese and four normal weight) via CD8+/CD4+ peripheral blood mononuclear cell RNA-sequencing and paired serum metabolomics, which showed differential kynurenine and lipid metabolism, which corresponded with altered T-cell transcriptome in obese populations. Overall, obesity is associated with differential transcriptional and metabolic programs in various disease sites. Further, these alterations, such as kynurenine and tryptophan metabolism, which impact both metabolism and immune phenotype, vary with sex and obesity together. This study warrants further in-depth investigation into obesity and sex-related alterations in cancers that may better define biomarkers of response to immunotherapy.

## 1. Introduction

Nearly one-third of the world’s population is classified as overweight or obese by the Body Mass Index (BMI), which characterizes obesity as a global health crisis [1,2,3]. The rates of obesity have been steadily climbing over time, with the number of overweight and obese individuals worldwide increasing from 857 million (20%) in 1980 to 2.1 billion (30%) in 2013 [4], and even higher to ~60% of the global population [1]. Amongst countries classified as higher-income or first-world nations, the most significant increases in adult obesity have occurred in the United States (~33%), Australia (~30%), and the United Kingdom (~25%) [4]. With this growing epidemic, there has been a vested interest in better understanding other health-related outcomes associated with obesity. Not surprisingly, obesity, especially morbid obesity (BMI > 40), has been identified as a risk factor for numerous medical conditions, including diabetes mellitus, hypertension, cardiovascular disease, metabolic syndrome, and some cancers [5]. Obesity is associated with an increased risk of 13 types of cancer [6], including, but not limited to breast, prostate, gastrointestinal (GI) (including colorectal, esophageal, liver, and pancreas), kidney, and multiple myeloma.

Recent investigations into understanding the impacts of obesity on tumor activity, the tumor microenvironment, and related immune components have led to the finding that obesity may improve survival in some cases [7]. This paradoxically protective action has subsequently been termed the “obesity paradox.” Interestingly, obese patients have been reported to respond better to treatment with anti-PD1 immune checkpoint inhibitors [8]. However, in seeking to identify the underlying biological etiology of the obesity paradox, we have realized this phenomenon may differ by sex. Findings of a study particularly interested in metastatic melanoma and the “obesity paradox” found the existence of the paradox is restricted to overweight/Class-I obesity, which is predominated by males with increased serum creatinine levels, a surrogate for skeletal muscle mass [9]. These same associations were not seen in females, who tend to have lower creatinine levels. Other findings have suggested that the protective features of the obesity paradox are rooted in alterations in transcriptional profiles of obese patients as compared to normal-weight patients [10,11] and alterations of the immune milieu [12]. Much of this literature highlights the fact that obesity impacts males and females differently. This is important to explore, given that globally, a higher proportion of women are obese than men [3], yet most obesity treatment and service options are not sex-informed [13]. Recently, the crosstalk between systemic and tumor metabolism has been identified as a modulator of cancer immunotherapy efficacy in patients [14]. Several clinical trials are now looking at ways to combine checkpoint inhibition with the modulation of various metabolic pathways to improve therapeutic outcomes. As sex affects both systemic and tumor metabolism, there is a demonstrated need to identify the influence of sex on tumor growth [15].

In this study, we aim to identify the unique tumor transcriptional and metabolic milieu associated with obesity, and sex in the context of obesity. To examine this phenomenon, we specifically queried patients with GI cancers (Colon (COAD) and Esophageal (ESCA) Adenocarcinomas), as these disease sites are both known to be impacted by obesity and sex differences. Further, these disease sites have been associated with the obesity paradox. Here, we highlight consistent alterations to transcriptional tumor metabolism of obese male and female patients, albeit to a different extent, in lipid and amino acid-associated metabolic pathways compared to their normal-weight counterparts. Additionally, there are varying metabolic pathway transcriptional differences between the male and female obese patient populations, associated with hormone metabolism. Further, we demonstrate how these translate to implications in immune-related pathways, such as IL-6 signaling. To support these findings, we utilized serum samples from obese and non-obese patients. We confirmed metabolic dysregulation that overlapped both transcriptionally and metabolically, and model pathways of interest to better examine patterns of predicted metabolic flux associated with both sex and obesity. 

## 2. Results

### 2.1. Obesity Drives Differential Transcriptomic Signature Associated with Metabolic Reprogramming in GI Cancers

Patients from the ORIEN database with an adenocarcinoma diagnosis and associated BMI data (n = 637, n = 244 CRC, Table 1) were classified as obese (BMI ≥ 30) or normal weight (18 ≤ BMI ≤ 25). Tumor RNA-sequencing data from the two divergent groups was compared and resulted in the identification of 2511 significantly down-regulated and 1518 significantly upregulated (adj *p* < 0.05 and |LogFC| > 0.58) transcripts (Figure 1A) in obese patients compared to normal weight patients. The most highly down-regulated transcripts, with decreased expression in obese populations, include CTRB1, CTRC, and CELA2B, all of which are chymotrypsin- and trypsinogen-related genes [16]. These pathways are responsible for digestion and digestive processes, which are biologically relevant to the function of the colon. Interestingly, and in relation to both metabolic and immune biology, when taken as a ratio of each other, these genes are associated with the resolution of inflammatory symptoms and promote speedier recovery of acute tissue injury [2]. Conversely, the top up-regulated transcripts, with increased expression in obese populations, include ACPP (phosphatases) [3], SCGB1D2 (glycoproteins) [4], and PGR (steroid hormone receptors) [5], which are all associated with hormonal regulation and metabolic processing. Gene set enrichment analysis (Normalize Enrichment Score > 2, q-value < 0.05; Figure 1B) enriched for several types of pathways, including translation, therapeutic resistance, metabolism, immune dysfunction, and cellular processes. Metabolic pipeline analysis (Figure 1C) enriched for several metabolic pathways associated with obesity in adenocarcinomas, including lipids (such as steroid hormone metabolism) and amino acids (such as tryptophan and kynurenine metabolism), both of which are associated with immune cell dysfunction.

### 2.2. Obesity Drives Differential Transcriptomic Signature in a Disease-Type-Specific Manner

Demographic tables of patients in the Colon Adenocarcinoma, TCGA-COAD (top), and Esophageal Adenocarcinoma TCGA-ESCA (bottom) cohorts revealed relatively similar distributions of race, tumor stage, and BMI by sex. Of note, the only statistically significant differences were in the number of males and females in TCGA ESCA, and differences in tumor stage (Table 2). RNA-sequencing data of obese and non-obese patient populations from TCGA Colon Adenocarcinoma (COAD) (Figure 1D) and TCGA Esophageal Adenocarcinoma (ESCA) (Figure 1E) were compared, and differential transcripts are reported in volcano plots. Venn diagrams demonstrated very little overlap of differentially expressed genes (DEGs) between the two GI datasets (Figure 1F), representing a non-significant overlap (*p* > 0.05, Hypergeometric test) between the two datasets. This clarified that while obesity is associated with a differential program in both COAD and ESCA, it varies by disease site. Further, gene set enrichment analysis of the DEGs in TCGA COAD (Figure 1G) and TCGA ESCA (Figure 1H) highlighted the enrichment of both metabolic and immune pathways in obese and non-obese samples. While the individual transcriptional changes were different between the two populations, enrichment of those transcripts resulted in similar types of pathways being altered. Further, pathway modeling showed differential transcription of metabolic pathways by disease site (Figure 1I; ESCA, left; COAD, right). This is demonstrated by the transcriptional model of tryptophan-NAD synthesis, where a majority of the downstream Nicotinamide Adenine Dinucleotide (NAD) components are down-regulated in obese ESCA populations, as compared to their normal weight counterparts, whereas there were large transcriptional increases in this pathway in the obese COAD populations, as compared to their normal weight counterparts. However, given a large enrichment of metabolic pathways on the transcriptional level, it was important to also assess how these transcriptional changes are reflected in the metabolome of obese patients.

### 2.3. Obesity Associated with Differential Lipid and Amino Acid-Related Metabolic Programs

To this end, we conducted metabolomics on patient serum samples to better understand the metabolic differences of obese GI adenocarcinoma patients. Using internal patient samples, given that metabolomics data are not available through the TCGA, we obtained pre-treatment serum samples from six gastrointestinal cancer patients (four normal weight and two overweight, by a BMI > 25). The demographic and clinical characteristics of these patients are summarized in Appendix A. We conducted global metabolomics using the Biocrates MxP Quant 500 kit, which measures up to 630 metabolites spanning 26 biochemical classes. This revealed distinct metabolic profiles that segregated patients by weight via Principal Component Analysis (PCA, Figure 2A). Unsurprisingly, we found this differential clustering to be driven by a large number of increased lipids in the obese population compared to the normal-weight population, as can be seen on the right side of the volcano plot (Figure 2B). In fact, there are clear, statistically significant differences in metabolite abundance when comparing obese (blue) to non-obese (purple) patients. The top 10 differential metabolites were all classified as lipids and were more abundant in the overweight patients as compared to the normal-weight patients (Figure 2C). In particular, metabolite set enrichment analysis (Figure 2D) revealed a large amount of lipid enrichment as compared to any other class of metabolites (e.g., amino acids, carbohydrates, etc.). This enrichment is largely attributed to variation in a multitude of lipids, including Ceramides, Triacylglycerols, Sphingomyelins, and Phosphatidylcholines. Interestingly, and concordantly with the TCGA RNA-sequencing data, many of these lipids are highly associated with immune reprogramming and dysfunction [6,7]. Given our previous enrichment for immune dysregulation comparing obese to non-obese patient populations, we felt it is necessary to additionally investigate the immune profiles of these patients.

### 2.4. Obesity Drives Differential CD8+ and CD1+4 Metabolic and Immune Transcriptional Profiles Regardless of Sex

To study the immune cell populations in the same six patients (four normal and two overweight), we conducted RNA-sequencing of flow cytometry-sorted CD8+ and CD14+ immune populations from peripheral blood mononuclear cells (PBMCs). This data revealed distinct transcriptional profiles in non-obese and obese populations for both the CD8+ (Figure 3A) and CD14+ (Figure 3B) cells, with more notable differences in the CD8+ cells. Gene set enrichment analysis of the DEGs was enriched for differences in immune pathways, not surprisingly, given these were sorted immune populations, as well as metabolism/metabolic pathways. One such pathway of interest was prostaglandin metabolism (Figure 3C, Appendix A), which was highly dysregulated in CD14+ cells of obese patients as compared to non-obese patients. This further revealed increased transcriptional production of Arachidonic Acid and Prostaglandin H2, and decreased production of HPETE metabolites and Leukotrienes in CD14+ cells of obese patients. Further, in CD8+ cells, prostanoid and thromboxane metabolism (Appendix A) were amongst some of the most highly dysregulated metabolic pathways, among other amino acid-related pathways, such as the urea cycle (Figure 3D). Urea cycle enrichment revealed modest increases in transcripts associated with ornithine generation (OAT), lactate and oxaloacetate production, and decreased argininosuccinate and arginine generation. Aside from metabolic enrichment, immune pathway enrichment was observed. Given the enrichment of immune pathways, predicted immunotherapy response (PD-L1 and CTLA-4) pathways were transcriptionally modeled in CD8+ and CD14+ cells (Figure 3E, Appendix A). Interestingly, this pathway was largely down-regulated in CD8+ cells. Conversely, this pathway was largely upregulated in CD14+ cells. This demonstrated that obesity may play a role in differential transcriptional reprogramming within different immune cell populations, which may specifically relate to altered immunotherapy response. 

### 2.5. Obese Males and Females Dysregulate Similar Amino Acid and Lipid-Related Metabolic Pathways to Varying Extents

While the differences between obese and non-obese populations are interesting, in terms of both transcriptional and metabolomic alterations, the sex differences associated with this phenomenon are largely of interest to us as well. It is well known that males have a higher incidence of GI cancers [8], females have significantly better overall survival with GI cancers [9], and in immunotherapy studies, women tend to mount more of an immune response [10]. Therefore, utilizing RNA-sequencing data from The Cancer Genome Atlas (TCGA), esophageal adenocarcinoma (ESCA) patient populations were designated as either obese (BMI ≥ 30) or non-obese (18 ≤ BMI ≤ 25), and then later split by sex, resulting in four groups. Similar analyses were conducted in TCGA colon adenocarcinoma (COAD) (Appendix A). Differential expression analysis (adjusted *p* < 0.05; |log2 Fold Change| > 0.58) revealed 5558 DEGs between obese and non-obese males (Figure 4A) and 480 DEGs between obese and non-obese females (Figure 4B). While there was extensive overlap (40 upregulated genes, 150 down-regulated genes) in the transcriptional dysregulation that occurs between males and females, there were notable differences transcriptionally. Consequently, there was also overlap in the types of pathways enriched in the obese populations, as compared to their non-obese counterparts for both males and females. Obese males as compared to non-obese males resulted in enrichment of pathways associated with stemness, differentiation, epigenetics, and metabolism (Figure 4C). Comparisons of obese females with non-obese females highlighted the enrichment of pathways associated with epithelial-to-mesenchymal transition, development, and metabolism (Figure 4D). Given that GSEA revealed both male and female obese patients’ dysregulated metabolism when compared to their non-obese counterparts, we applied our metabolic pipeline to understand which metabolic pathways were the most highly and statistically significant dysregulated in each of the two groups (Figure 4E). This analysis revealed alterations to both lipid and amino acid metabolism, some of which were similarly dysregulated in both sexes and some that were dysregulated uniquely in a single sex. One such pathway that is commonly significantly dysregulated in both sexes was tryptophan/kynurenine metabolism (Figure 4F; male, left; female, right). However, this immune-related pathway was dysregulated differently in each of the sexes. It was significantly transcriptionally down-regulated in females, with predicted decreases in the production of kynurenine from tryptophan. Conversely, in males, the earlier transcripts within this pathway were upregulated, indicating increased production of kynurenine from tryptophan. Taken together, this highlights the importance of not only considering obesity, but also sex, when assessing novel points of therapeutic leverage in GI cancer patients.

## 3. Discussion

Overall, this assessment of the obese tumor transcriptome compared to the non-obese transcriptome demonstrated a large amount of transcriptional metabolic dysregulation, which largely corresponds with serum-level microenvironmental metabolic dysregulation. In this study, we first assessed consortium GI adenocarcinoma transcriptomic sequencing datasets from ORIEN and TCGA (Appendix A). Comparing obese and non-obese individuals, and broken down further by sex, these datasets were then scrutinized via DEG, GSEA, and metabolic pipeline analysis to understand and predict metabolic dysregulation associated with obesity. To confirm findings from these large consortia data, sorted CD8+, CD14+ PBMCs and serum of patients treated at our institute were utilized for paired metabolomics (serum) and transcriptomics (PBMCs) to reveal the convergence of metabolic dysregulation both in the context of obesity and sex (Appendix A). More specifically, obesity is associated with differential transcriptome profiles in obese GI patient populations (Figure 1) regardless of sex, many of which were immune or metabolically altered. Similar to previously published studies, obesity corresponded with differential metabolomics in the serum microenvironment, largely differentiated from normal-weight patient serum by lipids [17,18,19,20,21] and amino acids, which are heavily associated with immune reprogramming (Figure 2). Further, obesity is associated with differential metabolic and immune transcriptional profiles in immune cells, which may correspond with the response to immunotherapy (Figure 3). Our findings are similar to the recent study performed by Ringel et al. that showed tumor-infiltrating lymphocytes obtained from a high-fat-diet (HFD) induced obese murine tumors enriched for several pathways related to fat and cholesterol metabolism including glycosphingolipid biosynthesis, steroid biosynthesis, and fatty acid metabolism [22]. Our study, in addition to the Ringel et al. [22] study provides evidence that systemic metabolism influences tumor metabolism. 

In our study, metabolic dysregulation trends in males and females were similar, with a much more prominent effect in females than in males (Figure 4). Clinically, it has been demonstrated that immune checkpoint inhibitors work better in obese cancer patients compared to normal-weight patients; however, this association is mainly seen in obese male patients [23]. Our prior collaborative work has confirmed the clinical phenomenon and further showed that HFD-induced obese male mice had significantly increased benefit from anti-PD-1 checkpoint inhibitors compared to diet-induced obese female mice or the lean male and female mice [24], necessitating a need to understand sex-related differences [22,25,26,27]. This, therefore, highlights the need to study this phenomenon in greater depth, especially with the integration of additional levels of data (e.g., paired tumor transcriptomics and metabolomics, circulating cytokines). 

While we believe this study provides vital information and evidence that obesity is associated with a differential metabolic and immune profile in both the tumor and systemic microenvironment of patients, we recognize that this study has flaws. The largest flaw is the use of the BMI as a measure of adiposity. The BMI fails to distinguish fat adequately from fat-free mass, such as muscle and bone and other bodily tissue [11]. Therefore, for more direct measures of fat, DEXA scans, and visceral adiposity measurements from imaging would better quantitate obesity. However, we are currently leveraging the wealth of publicly available data, where the most commonly used measure of obesity is the BMI. Additionally, we drew on transcriptomic data, conducting metabolism-specific transcriptional analysis as a surrogate for metabolomics. While we have evidence that the pipeline is accurate in predicting metabolic dysregulation [28,29], we know that metabolomics is the best possible read-out for metabolic dysregulation. Unfortunately, the publicly available metabolomics data are limited [30] and, further, are unavailable for most TCGA patients. Therefore, we utilized patient serum samples available to us, with available patient BMI information to ensure that we accurately captured obesity-associated effects. Here, we found that serum metabolomics from our patients corresponded with TCGA tumor transcriptomics. However, it is essential to note that we did have a limited sample size to draw on for this study. In order to enact clinical change, we plan to incorporate more patient samples in future studies. 

This is a very clinically significant issue, given the current global obesity epidemic. Future studies should, additionally, focus on other immune cell populations (beyond CD8+ and CD14+ cells), which may also play a role in carcinogenesis and therapeutic response, such as macrophages [31], which are impacted by visceral adiposity [32]. Understanding how obesity influences tumor metabolism and immunity may have therapeutic implications. Tumor metabolism has been targeted as a mode of cancer therapy for a long time, starting from chemotherapeutic agents such as 5-fluorouracil; however, there is a new interest these days in harnessing metabolic pathways to make immunotherapy more efficacious [33]. Therefore, in the future, we plan to assess not only the transcriptional and metabolic alterations that occur in the tumor and specific immune cell populations but also within the entirety of the immune milieu, both internal to and external to the tumor, in the context of both sex and obesity.

## 4. Methods

### 4.1. Study Design

The Oncology Research Information Exchange Network (ORIEN) is an alliance of 18 U.S. cancer centers established in 2014. All ORIEN alliance members utilize a standard Total Cancer Care^®^ (TCC) protocol. As part of the TCC study, participants agree to have their clinical data followed over time, to undergo germline and tumor sequencing, and to be contacted in the future by their provider if an appropriate clinical trial or other study becomes available https://www.orientcc.org [34]. TCC is a prospective cohort study with a subset of patients enrolled to the ORIEN Avatar program, which includes research use only (RUO) grade whole-exome tumor sequencing, RNA sequencing, germline sequencing, and collection of deep longitudinal clinical data with lifetime follow-up. Nationally, over 325,000 participants have enrolled in TCC. ASTER INSIGHTS, the commercial and operational partner of ORIEN, harmonizes all abstracted clinical data elements and molecular sequencing files into a standardized, structured format to enable aggregation of de-identified data for sharing across the Network. ORIEN Avatar patients were utilized if they were diagnosed with adenocarcinoma and consented to the TCC protocol from the participating members of ORIEN. 

### 4.2. Sequencing Methods 

ORIEN Avatar specimens undergo nucleic acid extraction and sequencing at HudsonAlpha (Huntsville, AL, USA) or Fulgent Genetics (Temple City, CA, USA). For frozen and OCT tissue DNA extraction, Qiagen QIASymphony DNA purification is performed, generating 213 bp average insert size. For frozen and OCT tissue RNA extraction, Qiagen RNAeasy plus mini kit is performed, generating 216 bp average insert size. For FFPE tissue, Covaris Ultrasonication FFPE DNA/RNA kit is utilized to extract both DNA and RNA, generating 165 bp average insert size. For DNA sequencing, preparation of ASTER INSIGHTS Whole Exome Sequencing (WES) libraries involves hybrid capture using an enhanced IDT WES kit (38.7 Mb) with additional custom-designed probes for double coverage of 440 cancer genes. Library hybridization is performed at either single or 8-plex, and sequenced on an Illumina NovaSeq 6000 instrument generating 100 bp paired reads. WES is performed on tumor/normal matched samples with the normal covered at 100× and the tumor covered at 300× (additional 440 cancer genes covered at double coverage; 200× for normal and 600× for tumor). Both tumor/normal concordance and gender identity QC checks are performed. The minimum threshold for hybrid selection is >80% of bases with >100× fold coverage for tumor and >50× fold coverage for normal. RNA sequencing (RNAseq) is performed using the Illumina TruSeq RNA Exome with single library hybridization, cDNA synthesis, library preparation, sequencing (100 bp paired reads at Hudson Alpha, 150 bp paired reads at Fulgent) to a coverage of 100 M total reads/50 M paired reads.

### 4.3. The Cancer Genome Atlas Transcriptional Analysis 

Raw feature counts were downloaded from the TCGA, normalized, and differential expression analysis was conducted using DESeq2 [34]. Differential expression rank order was utilized for subsequent Gene Set Enrichment Analysis (GSEA), performed using the clusterProfiler package in R. Gene sets queried included the Hallmark, Canonical pathways, and GO Biological Processes Ontology collections available through the Molecular Signatures Database (MSigDB) [35].

### 4.4. Metabolic Pipeline Analysis

The metabolic pipeline utilizes Differentially Expressed Gene output. For differential gene expression analysis of raw counts, RNA-seq counts were processed to remove genes lacking expression in more than 80% of samples, and the DESeq2 package was utilized for differential analysis. However, most datasets were downloaded as normalized counts. In these cases, the “contrast matrix” function of the Limma package in R, was utilized to obtain differential analysis. Euclidian distances and hierarchal clustering were utilized to determine sample similarity. Scores for each gene were produced by multiplying the −log (adjusted *p* value) *logFC on a gene level. Further, absolute value scores were produced by taking the absolute value of the scores for each gene. Using the previously published pipeline [14], we assessed transcriptional metabolic pathway dysregulation in several datasets.

### 4.5. Metabolomics—Biocrates Assays

Serum samples were prepared and analyzed in the Roswell Park Comprehensive Cancer Center Bioanalytics, Metabolomics, and Pharmacokinetics Shared Resource, using the MxP Quant 500 kit (Biocrates Life Sciences AG, Innsbruck, Austria) in accordance with the user manual. 10 μL of each supernatant, quality control (QC) samples, blank, zero sample, or calibration standard were added on the filterspot (already containing internal standard) in the appropriate wells of the 96-well plate. The plate was then dried under a gentle stream of nitrogen. The samples were derivatized with phenyl isothiocyanate (PITC) for the amino acids and biogenic amines and dried again. Sample extract elution was performed with 5 mM ammonium acetate in methanol. Sample extracts were diluted with either water for the HPLC-MS/MS analysis (1:1) or kit running solvent (Biocrates Life Sciences AG) for flow injection analysis (FIA)-MS/MS (50:1), using a Sciex 5500 mass spectrometer. Data were processed using MetIDQ software (Biocrates Life Sciences AG, vsn Oxygen-DB110-3005), and Limma (vsn 3.56.2) [36] for differential metabolite analysis.

### 4.6. Modeling with Cytoscape

Pathway maps were generated using Cytoscape software (vsn 3.8.2) [37], specifically the VizMapper functions. Pathway maps were adapted from existing pathway maps in WikiPathways (vsn 3.3.7) [38]. The DESeq2 output for DEG analysis and Limma output for metabolomics analysis were utilized to direct shading of genes and metabolites within the pathway: red (positive fold change, statistically significant), blue (negative fold change, statistically significant), or gray (non-statistically significant), for individual cancer sites transcripts (triangles) and metabolites (rounded rectangles).

## Figures and Tables

**Figure 1 ijms-24-10847-f001:**
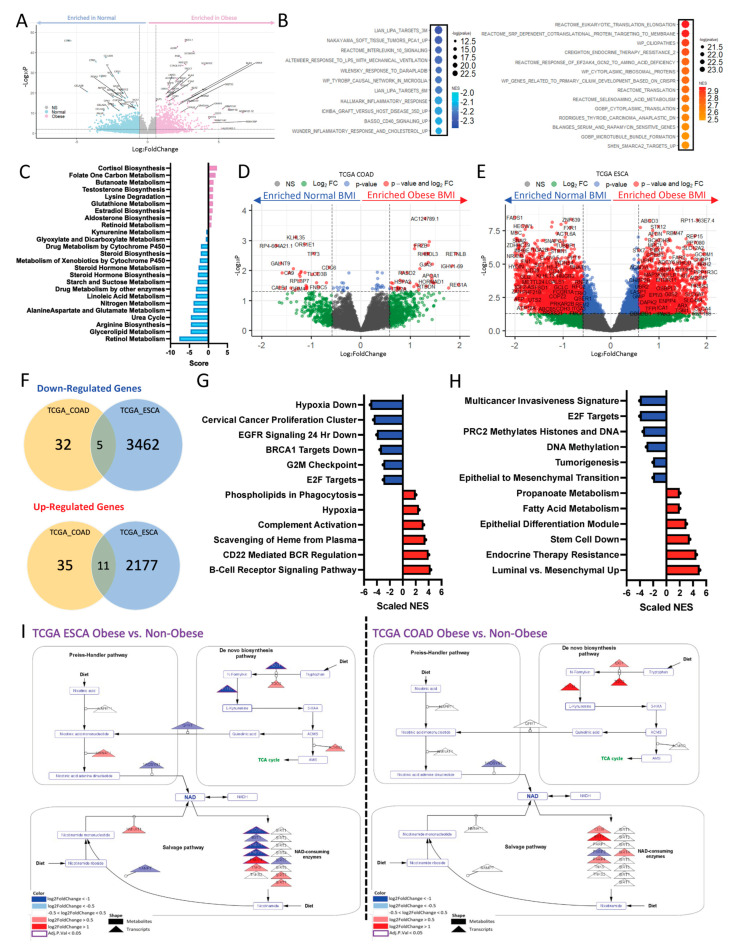
Obesity corresponds with differential tumor transcriptomic signatures associated with metabolic reprogramming. (**A**) Volcano plot showing significantly down-regulated transcripts (blue, adj. *p* < 0.05 and |logFC| > 1.5) and up-regulated (pink, adj. *p* < 0.05 and |logFC| > 1.5) transcripts in ORIEN Adenocarcinoma patients that were obese (BMI ≥ 30), as compared to normal weight (BMI ≤ 25) patients. (**B**) Gene set enrichment analysis (GSEA) of these differentially expressed genes from obese vs. non-obese ORIEN analysis show pathways down-regulated (blue, left) and up-regulated (red, right) in obese patients, as compared to normal patients, which enrich for metabolic pathways, immune pathways and biological processes. (**C**) Further transcriptional pathway analysis specific to metabolic pathways in these same patients, enriched for several lipid and amino acid specific metabolic pathway dysregulation. Similar analysis comparing the transcriptome of obese patients to non-obese patients were conducted and displayed as volcano plots for the (**D**) TCGA colon adenocarcinoma (COAD) and (**E**) TCGA esophageal carcinoma (ESCA), where ESCA displayed far more significantly differentially expressed genes (red, adj. *p* < 0.05 and |logFC| > 1.5) than COAD. To demonstrate these transcriptional alterations, occur in a site-specific manner, (**F**) we looked for overlap between the significantly down-regulated (top) and up-regulated (bottom) genes between the COAD and ESCA datasets, and found minimal overlap. Gene set enrichment analysis of (**G**) COAD and (**H**) ESCA revealed transcriptional enrichment of cellular processes decreased in obese patients (blue) like G2M checkpoint, and methylation, and up-regulated processes (red) like immune complement activation and therapeutic resistance. Further, metabolic analysis revealed several similar metabolic pathways to be differentially regulated in both ESCA and COAD, albeit to a different extent. For example, (**I**) Modeling of the tryptophan/kynurenine/NAD pathway reveals a majority of transcripts are down-regulated (blue triangles, left), in ESCA obese as compared to normal patients. Conversely, a majority of transcripts are up-regulated (red triangles, right), in COAD obese as compared to normal patients.

**Figure 2 ijms-24-10847-f002:**
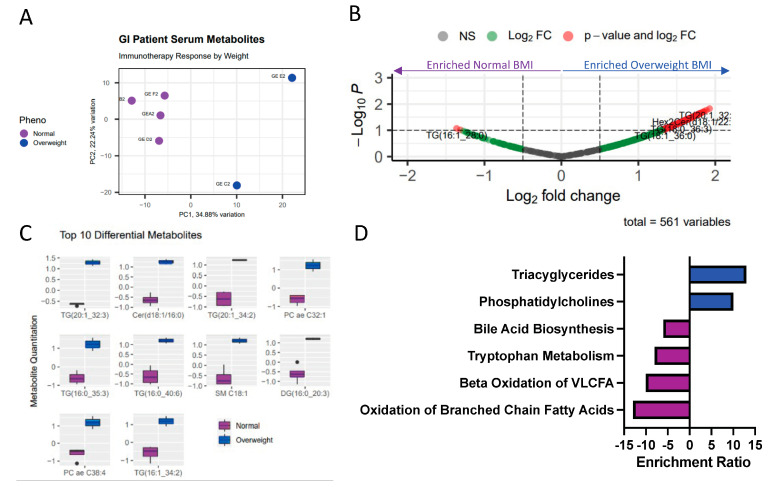
Obesity associated with differential circulating serum metabolome. The Biocrates MxP Quant500 metabolomics assay was utilized to assess the metabolome of GI patient serum samples of normal weight (purple, n = 4) and overweight (blue, n = 2) patients. (**A**) Principal component analysis reveals distinction between normal weight (purple) and overweight (blue) serum samples, based on the abundance of 630 metabolites measured by metabolomics study. (**B**) Volcano plot shows significantly differential metabolite abundances (*p* < 0.1, |logFC| > 1.5) between the normal BMI and overweight BMI patient groups. (**C**) Boxplots quantify the top 10 significantly differentially abundant metabolites, all of which are lipids from 4 different biochemical classes. (**D**) Corresponding with the top 10 differentially abundant metabolites, the metabolite set enrichment of all significantly differentially abundant metabolites highlighted alterations to lipid and amino acid pathways. This enrichment, specifically, indicated decreased lipids, tryptophan metabolites and bile acid metabolites in the obese patients, as compared to the non-obese patients (purple), and phosphatidylcholines were increased in obese patients, as compared to normal weight patient serum samples.

**Figure 3 ijms-24-10847-f003:**
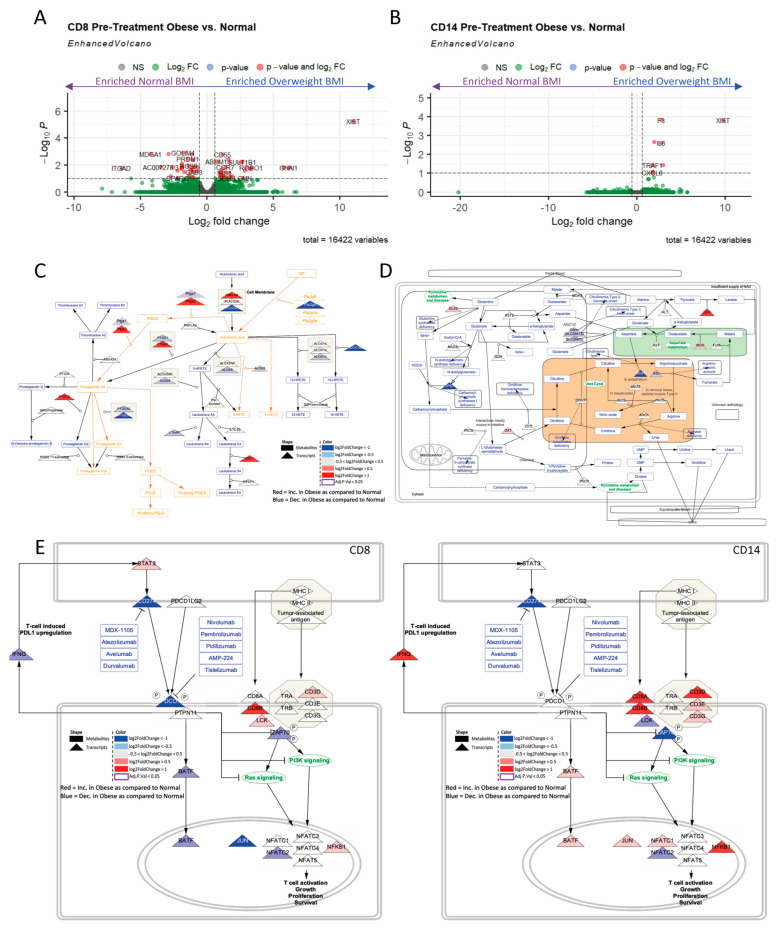
Obesity associated with differential immune cell transcriptome. CD8+ and CD14+ PBMCs sorted from matched patient samples (Figure 2) with metabolomics (obese n = 2, normal weight n = 4) and utilized for RNA-sequencing. Obesity is associated with differential transcriptional profile in both (**A**) CD8+ and (**B**) CD14+ cells of patients, with a larger number of significantly differentially expressed genes (red, adj. *p* < 0.05 and |logFC| > 1.5) in the CD8+ cells. (**C**) Prostaglandin pathway is highly upregulated transcriptionally (red triangles), largely driven by upregulation of PTGS2, in the Obese patients as compared to the non-obese patient’s CD14+ sorted T-cells. (**D**) The urea pathway is highly upregulated transcriptionally in the Obese patients as compared to the non-obese patient’s CD8+ sorted T-cells, largely driven by LDH, transcriptionally. (**E**) The PD-L1 response pathway is differentially dysregulated in CD8+ (left) and CD14+ (right) cells of obese as compared to non-obese patients, with a large number of decreased (blue triangle) transcripts in the CD8+ cells associated with obesity as compared to normal weight patients. Whereas, in CD14+ cells, there are a large number of increased (red triangle) transcripts associated with obesity, as compared to normal weight samples.

**Figure 4 ijms-24-10847-f004:**
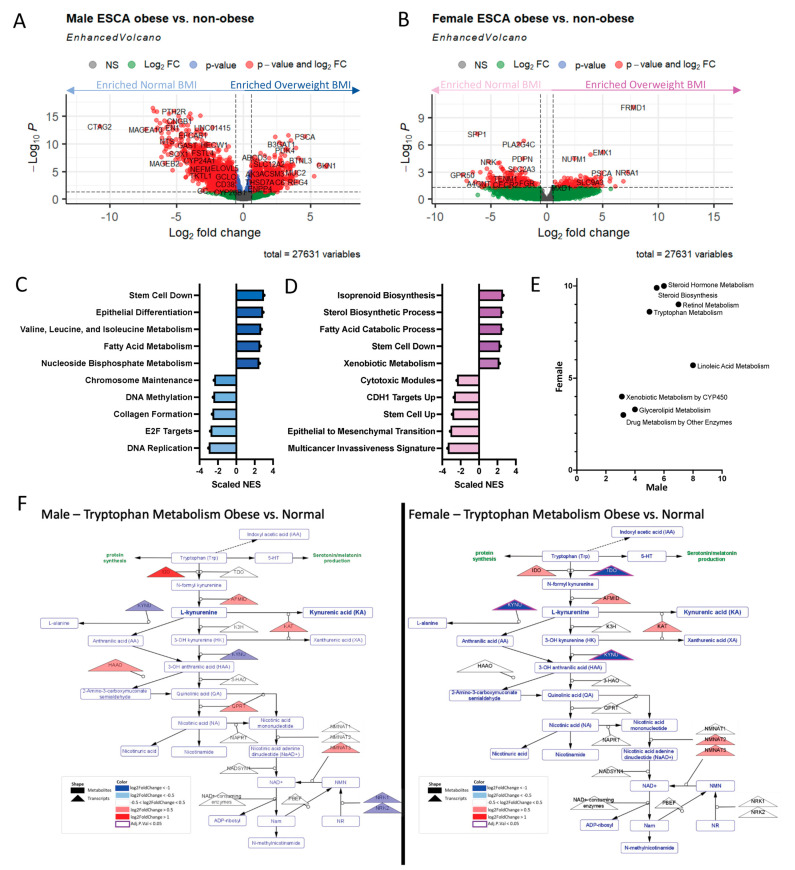
Gender induces differential transcriptional metabolic profile. TCGA datasets were broken into males and females. Differential gene expression was assessed comparing obese and non-obese patient sets within each sex. (**A**) Male ESCA assessment resulted in a large number of differentially expressed genes DEGs; adj. *p* < 0.05, |logFC| > 1.5, red) when comparing patients with BMI considered normal (left, light blue) to those considered overweight (right, dark blue). (**B**) Female ESCA assessment resulted in a large number of differentially expressed genes (adj. *p* < 0.05, |logFC| > 1.5, red) when comparing patients with BMI considered normal (left, light pink) to those considered overweight (right, dark pink). (**C**) Gene set enrichment analysis (GSEA) of male ESCA DEGs resulted in enrichment of cellular processes, metabolism, and epigenetic associated pathways. (**D**) Gene set enrichment analysis (GSEA) of female ESCA DEGs resulted in enrichment of similar cellular processes, and metabolism associated pathways. (**E**) Metabolic pipeline analysis resulted in enrichment of steroid, tryptophan, and other lipid metabolic pathways in males and females. (**F**) Comparison of the transcriptomics profiles of obese vs. non-obese patients, specifically the tryptophan metabolic pathway, in males (left) and females (right) demonstrated dysregulation of transcripts (triangles) in obese vs. normal weight patients, albeit to a different extent. Many of the transcripts were largely upregulated with obesity (red) in males and downregulated with obesity (blue) in females.

**Table 1 ijms-24-10847-t001:** Demographic and clinical characteristics of colorectal cancer patients from ORIEN database.

	ORIEN CRC		
Variable	Age > 50	Age <= 50	*p*
	n = 364 (n%)	n = 96 (n%)	
Race			0.621
White	313 (86)	80 (83)	
Non-white	51(14)	16 (17)	
Not reported	0 (0)	0 (0)	
Gender			0.801
Female	163 (45)	45 (47)	
Male	201 (55)	51 (53)	
Tumor stage			0.009
I (IA, IB)	21 (5)	3 (3)	
II (IIA/B/C)	90 (24)	10 (10)	
III (IIIA/B/C)+	104 (29)	34 (35)	
IV (IVA/B/C)+	87 (24)	31 (32)	
Not reported	62 (17)	18 (18)	
BMI			0.624
BMI < 18	4 (1)	1 (1)	
BMI 18–24.9	70 (19)	24 (25)	
BMI 24.9–30	111 (30)	25 (26)	
BMI >= 30	109 (30)	28 (29)	
Missing	70 (19)	18 (18)	

**Table 2 ijms-24-10847-t002:** Demographic tables of patients in the Colon Adenocarcinoma, TCGA-COAD (top), and Esophageal Adenocarcinoma TCGA-ESCA (bottom) cohorts.

	Characteristics by Gender in TCGA-COAD, 438 Cases			
Variable	Levels	Female (n = 208)	Males (n = 230)	*p*
**race**	asian	2 (1)	5 (2)	0.4
	black or african american	31 (15)	28 (12)	
	not reported	71 (34)	92 (40)	
	white	104 (50)	1005 (46)	
**tumor stage**	I (IA, IB)	33 (16)	42 (18)	0.6
	II (IIA, IIB)	70 (34)	66 (29)	
	III (IIIA, /B/B)+	100 (48)	115 (50)	
	no report	5 (2)	7 (3)	
**bmicat2**	BMI >= 30	41 (36)	37 (33)	0.3
	BMI 18–24.9	39 (34)	32 (28)	
	BMI 24.9–30	34 (30)	44 (39)	
		**median (range)**		
**Age at initial diagnosis**		67 (34–90)	68 (31–90)	0.2
	**Characteristics by Gender in TCGA-ESCA, 87 Cases**			
**Variable**	**Levels**	**Female (n = 13)**	**Males (n = 74)**	** *p* **
**race**	asian	0 (0)	1 (1)	0.9
	black or african american	0 (0)	0 (0)	
	not reported	2 (15)	14 (19)	
	white	11 (85)	59 (80)	
**tumor stage**	I (IA,IB)	6 (46)	7 (9)	0.007
	II (IIA,IIB)	4 (31)	20 (27)	
	III (IIIA,/B/B)+	2 (15)	31 (42)	
	no report	1 (8)	16 (22)	
**bmicat2**	BMI >= 30	4 (33)	19 (27)	0.8
	BMI 18–24.9	4 (33)	31 (44)	
	BMI 24.9–30	4 (33)	44 (39)	
		**median (range)**		
**Age at initial diagnosis**		74 (28–84)	65 (28–84)	0.03

## Data Availability

All data generated and/or analyzed in this study are either available on the Gene Expression Omnibus (GSE provided) or are available from the corresponding author upon reasonable request.

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
