# Peer review of "Metabolic Dysregulation Explains the Diverse Impacts of Obesity in Males and Females with Gastrointestinal Cancers"

_ijms, 2023, doi:10.3390/ijms241310847_

Round 1

Reviewer 1 Report

The manuscript entitled “Metabolic dysregulation explains the diverse impacts of obesity in males and females with gastrointestinal cancers” investigated the transcriptional and metabolic differences associated with obesity in adenocarcinomas of the gastrointestinal (GI) tract, with a focus on the potential impact of sex on these alterations. While the integration of multiple data sources is a strength of this study, it would be beneficial to validate the key findings via more specific assays with the patients’ samples. The authors performed metabolomics of serum samples and RNA-seq of CD8+ and CD14+ cells in PBMCs using samples from 4 normal and 2 overweight patients. It is important to verify their high-throughput results with more specific assays, such as qPCR of the top DEGs associated with corresponding metabolic changes. The demographic and clinical information of these 6 patients should also be provided for a more comprehensive interpretation of the results.

Minor comment:

In the first sentence of “Methods” in abstract: “gastrointestinal (GI) track” should be “gastrointestinal (GI) tract”.

None

Author Response

The manuscript entitled “Metabolic dysregulation explains the diverse impacts of obesity in males and females with gastrointestinal cancers” investigated the transcriptional and metabolic differences associated with obesity in adenocarcinomas of the gastrointestinal (GI) tract, with a focus on the potential impact of sex on these alterations. While the integration of multiple data sources is a strength of this study, it would be beneficial to validate the key findings via more specific assays with the patients’ samples. The authors performed metabolomics of serum samples and RNA-seq of CD8+ and CD14+ cells in PBMCs using samples from 4 normal and 2 overweight patients. It is important to verify their high-throughput results with more specific assays, such as qPCR of the top DEGs associated with corresponding metabolic changes. The demographic and clinical information of these 6 patients should also be provided for a more comprehensive interpretation of the results.

  • We would like to thank the reviewer for the insightful comments. We have added the table below as a supplementary table that describes the demographic and clinical information of the patients.

Age

Median/Min/Max

67/46/79

Body Mass Index (BMI)

Median/Min/Max

24.3/21/3/30.7

Gender

Male

5 (83.0%)

Female

1 (17.0%)

Immune Checkpoint

Inhibitor

Pembrolizumab

4 (67.0%)

Nivolumab

2 (33.0%)

Race

Black

1 (17%)

White

5 (83%)

Tumor location

Gastroesophageal junction

3 (50%)

Esophageal

2 (33.3%)

Gastric

1 17.0%)

Sigmoid colon

1 (8.3%)

Histology

Adenocarcinoma

5 (83%)

Squamous cell Cancer

1 (8.3%)

We also agree with the reviewer that it is important to verify the high-throughput results with more specific assays; however, unfortunately, this could not be performed due to limited patient samples. We are planning to conduct confirmatory studies to confirm our findings.

Minor comment: In the first sentence of “Methods” in abstract: “gastrointestinal (GI) track” should be “gastrointestinal (GI) tract”.

  • Thank you for the suggestion. We have corrected it.

Reviewer 2 Report

The author aims to identify the unique tumor-associated transcriptional and metabolic milieu associated with obesity and with sex in the context of obesity. They highlight consistent alterations to transcriptional tumor metabolism of obese male and female patients, albeit to a different extent, in lipid and amino acid-associated metabolic pathways compared to their normal-weight counterparts. The author used serum samples from obese and non-obese patients, as well as publicly available metabolomics data that corresponds with The Cancer Genome Atlas (TCGA) samples. They confirmed metabolic dysregulation that overlapped both transcriptionally and metabolically and model pathways of interest to better examine patterns of metabolic flux associated with both sex and obesity. Albeit, I consider these findings to provide new insight into cancer-related fields, I still have some suggestions.

1, Most figures are highly professional; however, the authors should guide the readers to the meaning of the images appropriately; otherwise, it will likely cause misunderstandings. Therefore, I suggest the author consider revising these figure legends again.

2, In Figure 1F, the author used Venn diagrams to demonstrate very little overlap of differentially expressed genes between the two TCGA GI datasets. So far, the tumor infiltrates immune cells and is vital for patient survival. Therefore, it is worth exploring these genes correlated with immune cells by using the "TIMER" (http://timer.cistrome.org) analysis tool (PMID: 32442275, 34329194).

3, The author found alterations to steroid metabolism and tryptophan/kynurenine metabolism in obese patients, both of which are highly associated with disease severity and immune cell dysfunction. These findings were further confirmed using the TCGA colorectal adenocarcinoma (CRC) and esophageal adenocarcinoma (ESCA) data,  However, it would be fascinating if these data could be correlated with other clinical databases. Therefore, I suggest the authors can validate their data via Proteinatlas or cBioportal, and discuss these methodologies and literature as well as the validated data for cancer recurrence or metastasis in the manuscript (PMID: 17008526, 22588877, 32064155). 

4, This study warrants further in-depth investigation into obesity and sex-related alterations in cancers that may better define biomarkers of response to immunotherapy. However, it would be much better if the authors could provide some Workflow or Scheme for this research, I suggest that they can take a look at the recent paper in MDPI (PMID: 35565469) 

5, There are few typo issues for the authors to pay attention to; Please also unify the writing of scientific terms. “Italic, capital”? Meanwhile, the font is too small for some of the current figures.

Author Response

The author aims to identify the unique tumor-associated transcriptional and metabolic milieu associated with obesity and with sex in the context of obesity. They highlight consistent alterations to transcriptional tumor metabolism of obese male and female patients, albeit to a different extent, in lipid and amino acid-associated metabolic pathways compared to their normal-weight counterparts. The author used serum samples from obese and non-obese patients, as well as publicly available metabolomics data that corresponds with The Cancer Genome Atlas (TCGA) samples. They confirmed metabolic dysregulation that overlapped both transcriptionally and metabolically and model pathways of interest to better examine patterns of metabolic flux associated with both sex and obesity. Albeit, I consider these findings to provide new insight into cancer-related fields, I still have some suggestions.

1, Most figures are highly professional; however, the authors should guide the readers to the meaning of the images appropriately; otherwise, it will likely cause misunderstandings. Therefore, I suggest the author consider revising these figure legends again.

Thank you for these comments, we have revised our figure legends to improve clarity and direct the readers to better understand the purpose of each figure. We agree that this has improved the focus of the paper, and will guide the ease of understanding.

2, In Figure 1F, the author used Venn diagrams to demonstrate very little overlap of differentially expressed genes between the two TCGA GI datasets. So far, the tumor infiltrates immune cells and is vital for patient survival. Therefore, it is worth exploring these genes correlated with immune cells by using the "TIMER" (http://timer.cistrome.org) analysis tool (PMID: 32442275, 34329194).

We agree that this is interesting, and to answer a different question, we did previously assess the RNA-sequencing data using 6 different deconvolution software (MCPCounter, XCell, TIMER, EPIC, Cibersort, and Quantiseq). However, for the purposes of this paper, we were interested in the circulating immune cells, and the immune milieu (microenvironment) created within the context of obesity. While we think the immune deconvolution is an important question, it doesn’t address the particular question we wanted to ask.

3, The author found alterations to steroid metabolism and tryptophan/kynurenine metabolism in obese patients, both of which are highly associated with disease severity and immune cell dysfunction. These findings were further confirmed using the TCGA colorectal adenocarcinoma (CRC) and esophageal adenocarcinoma (ESCA) data. However, it would be fascinating if these data could be correlated with other clinical databases. Therefore, I suggest the authors can validate their data via Proteinatlas or cBioportal and discuss these methodologies and literature as well as the validated data for cancer recurrence or metastasis in the manuscript (PMID: 17008526, 22588877, 32064155).  

Thank you for this comment. We agree this would be very interesting. There was a good amount of correlation between the ORIEN colon adenocarcinoma datasets and TCGA colon adenocarcinoma datasets, which functioned as our validation set in the confines of this study. We then utilized patient samples for metabolomics and matched transcriptomics, where we knew all of the clinical outcomes, to determine whether our predicted pathway dysregulation could be confirmed in on-site samples.

4, This study warrants further in-depth investigation into obesity and sex-related alterations in cancers that may better define biomarkers of response to immunotherapy. However, it would be much better if the authors could provide some Workflow or Scheme for this research, I suggest that they can take a look at the recent paper in MDPI (PMID: 35565469) 

We agree, this study does warrant further investigation into the interplay between obesity and sex-related alterations in cancer. We have now created a supplementary figure (Supp Figure 4) with our study workflow, to help aid in the understanding of this figure. Please see below:

5, There are few typo issues for the authors to pay attention to; Please also unify the writing of scientific terms. “Italic, capital”? Meanwhile, the font is too small for some of the current figures.

Thank you for this feedback, we have gone through to harmonize all the text for both clarity and consistency.

Reviewer 3 Report

The study “Metabolic dysregulation explains the diverse impacts of obesity in males and females with gastrointestinal cancers” by Rosario et al. is a somewhat descriptive analysis of the impact of obesity on various metabolic and transcriptional programs in GI cancers. The manuscript is well written, but has complicated language with long sentences that can be easily cut down. The figures are extremely poorly done though, and their quality does not meet the standards of the journal. As such, the manuscript needs a major revision before I can recommend it for publication.

Major Comments –

1.      All figures are extremely low in resolution and very poorly done. They should be easily legible, and none of them are. Following comments address each individual figure.

2.      The Volcano plots (Figure 1A,D and E), GSEA (Figure 1B) and pathway modelling (Figure 1 I) shown in figure 1 are extremely low resolution and impossible to read, even when blown up on a large screen. These MUST BE replaced by high quality figures. The purpose of volcano plots, in particular, is to be legible to spot differences. Its impossible to make out anything with the current low-resolution figure.

3.      Figure 2 needs to be enlarged. Whole figure has poor resolution. Please provide high quality figures.

4.      Figure 3 suffers from poor resolution as well. 3C, D and E are particularly illegible to me. Please replace with high quality pathway maps.

5.      Figure 4 should be enlarged. Figure 4F needs higher resolution.

6.      There are multiple typographical errors in the manuscript that need to be corrected. You may use a free tool like Grammarly that would help point out a lot of these.

7.      Multiple sentences in the MS are made extremely long with the use of commas. Please spot them and cut them down wherever possible.

1. Use of English language is fine but there are typographical errors.

2. Most sentences are overly complicated and made extremely long. They can be and should be cut down.

Author Response

The study “Metabolic dysregulation explains the diverse impacts of obesity in males and females with gastrointestinal cancers” by Rosario et al. is a somewhat descriptive analysis of the impact of obesity on various metabolic and transcriptional programs in GI cancers. The manuscript is well written but has complicated language with long sentences that can be easily cut down. The figures are extremely poorly done though, and their quality does not meet the standards of the journal. As such, the manuscript needs a major revision before I can recommend it for publication.

The majority of the comments from Reviewer 3 addressed figure quality. We appreciate these comments and realized that upon integration of our figures into the word document, we lost quality. To address this, we have removed the figures from the integrated document, and instead provided a PDF document with the compiled figures, and a word document with the figure legends. Given the importance of legible figures, we feel these were extremely important comments for us to address, and appreciate the effort this reviewer put into assessing our figures. All major comments have been addressed from this reviewer.

 Major Comments

  1. All figures are extremely low in resolution and very poorly done. They should be easily legible, and none of them are. Following comments address each individual figure.
  2. The Volcano plots (Figure 1A, D and E), GSEA (Figure 1B) and pathway modelling (Figure 1 I) shown in figure 1 are extremely low resolution and impossible to read, even when blown up on a large screen. These MUST BE replaced by high quality figures. The purpose of volcano plots, in particular, is to be legible to spot differences. It’s impossible to make out anything with the current low-resolution figure.
  3. Figure 2 needs to be enlarged. Whole figure has poor resolution. Please provide high quality figures.
  4. Figure 3 suffers from poor resolution as well. 3C, D and E are particularly illegible to me. Please replace with high quality pathway maps.
  5. Figure 4 should be enlarged. Figure 4F needs higher resolution.
  6. There are multiple typographical errors in the manuscript that need to be corrected. You may use a free tool like Grammarly that would help point out a lot of these.

Thank you for this feedback. We have gone through the manuscript in depth and made corrections to increase clarity, improve grammar, and reduce sentence length. We agree that this improves the manuscript vastly. They are all tracked changes within the manuscript now.

  1. Multiple sentences in the MS are made extremely long with the use of commas. Please spot them and cut them down wherever possible.

Thank you for this feedback. We have gone through the manuscript in depth and made corrections to increase clarity, improve grammar, and reduce sentence length. We agree that this improves the manuscript vastly. They are all tracked changes within the manuscript now.

Comments on the Quality of English Language

  1. Use of English language is fine but there are typographical errors.
  2. Most sentences are overly complicated and made extremely long. They can be and should be cut down.

- We agree with this feedback. Please see our response above.

Round 2

Reviewer 1 Report

Despite the lack of low-throughput validation of their findings through high-throughput data, the other revisions the authors made have improved the manuscript significantly, especially the inclusion of the clinical and demographic information.

Reviewer 3 Report

The authors have made a considerable effort to address my concerns and I am happy to accept the manuscript at this stage.